# In Vitro α-Amylase and α-Glucosidase Inhibitory Activity of Green Seaweed *Halimeda tuna* Extract from the Coast of Lhok Bubon, Aceh

**DOI:** 10.3390/plants12020393

**Published:** 2023-01-14

**Authors:** Mohamad Gazali, Odi Jolanda, Amir Husni, Fadzilah Adibah Abd Majid, Rina Syafitri

**Affiliations:** 1Department of Marine Science, Faculty of Fisheries and Marine Science, Teuku Umar University, Aceh 23681, Indonesia; 2Department of Fisheries, Faculty of Agriculture, Universitas Gadjah Mada, Yogyakarta 55281, Indonesia; 3Department of Aquatic Product Technology, Faculty of Fisheries and Marine Science, IPB University, Bogor 16680, Indonesia; 4Institute of Marine Biotechnology, Universiti Malaysia Terengganu, Kuala Terengganu 21030, Malaysia; 5Department of Fisheries, Faculty of Fisheries and Marine Science, Teuku Umar University, Aceh 23681, Indonesia; 6Department of Agribusiness, Faculty of Agriculture, Teuku Umar University, Aceh 23681, Indonesia

**Keywords:** *Halimeda tuna*, α-amylase, α-glucosidase, methanol extract, ethyl acetate fraction

## Abstract

Seaweed belongs to marine biota and contains nutrients and secondary metabolites beneficial for health. This study aimed to determine the antidiabetic activity of extracts and fractions of green seaweed *Halimeda tuna*. The *H. tuna* sample was extracted with the maceration method using methanol and then partitioned using ethyl acetate and water to obtain ethyl acetate and water fractions. The methanol extract, ethyl acetate fraction, and water fraction of *H. tuna* were tested for their inhibitory activity against α-amilase and α-glucosidase. The methanol extract and the fractions with the highest inhibitory activity were phytochemically tested and analyzed using gas chromatography–mass spectrometry (GC-MS). The results showed that the ethyl acetate fraction (IC_50_ = 0.88 ± 0.20 mg/mL) inhibited α-amylase relatively similar to acarbose (IC_50_ = 0.76 ± 0.04 mg/mL). The methanol extract (IC_50_ = 0.05 ± 0.01 mg/mL) and the ethyl acetate fraction (IC_50_ = 0.01 ± 0.00 mg/mL) demonstrated stronger inhibitory activity against α-glucosidase than acarbose (IC_50_ = 0.27 ± 0.13 mg/mL). Phytochemical testing showed that the methanol extract and the ethyl acetate fraction contained secondary metabolites: alkaloids, flavonoids, steroids, and phenol hydroquinone. The compounds in methanol extract predicted to have inhibitory activity against α-amylase and α-glucosidase were Docosanol, Neophytadiene, Stigmasta-7,22-dien-3-ol,acetate,(3.beta.,5.alpha.,22E), Octadecanoic acid,2-oxo-,methyl ester, and phytol, while those in the ethyl acetate fraction were n-Nonadecane, Phytol, Butyl ester, 14-.Beta.-H-pregna, Octadecenoic acid, and Oleic acid.

## 1. Introduction

Diabetes is a metabolic disease caused by low insulin production or insulin hormone not functioning properly, or it is caused by both [1]. It increases sugar accumulation in the blood. Blood glucose levels increasing above 200 mg/dL is an early symptom of diabetes mellitus (hyperglycemia) [2]. Data from the World Health Organization (WHO) in 2021 reported that people with diabetes increased from 108 million in 1980 to 422 million in 2014, with the majority living in low-to-middle-income countries. In 2019, diabetes ranked as the ninth cause of death. Consumption patterns can affect health, and several studies have shown a relationship between high consumption of calories or food and high glycemic index, and increased risk of type 2 diabetes mellitus [3].

One of the efforts to cure type 2 diabetes mellitus is using a therapeutic approach to inhibit the degradation of oligo- and disaccharides during digestion. This can reduce postprandial hyperglycemia by inhibiting enzymes that hydrolyze carbohydrates in the digestive tract [4], i.e., the α-amylase enzyme in saliva and pancreas and α-glucosidase enzymes located at the edge of the small intestine [5]. Synthetic α-glucosidase and α-amylase inhibitors, such as acarbose, have been widely used to treat type 2 diabetes patients. However, drug candidates such as acarbose, metformin, and voglibose have side effects, such as gastrointestinal disturbances, nausea and vomiting, hepatic impairment, and dizziness. Therefore, there is a need for safer and more effective antidiabetic drug candidates derived from plants, one of which is from seaweed.

Seaweed contains a high amount of antioxidants and can reduce hyperglycemia in diabetic patients [6]. Several in vitro and in vivo studies have demonstrated the significant function of polyphenols from green seaweeds in preventing and managing type 2 diabetes mellitus [7]. One type of green seaweed known to have potential as an antidiabetic is *H. tuna*. *H. tuna* is one of the green seaweed species reported to have biological activities, such as antimicrobial [8,9], antioxidant [8,10,11,12], antifungal [9], and antitumor activities, and inhibition of α-glucosidase [8].

Chlorophyta, Rhodophyta, and Phaeophyta seaweeds are almost evenly distributed in Aceh, Indonesia [13]. *Halimeda* sp. is one of the green seaweeds that are abundant and ecologically distributed along the coast of Lhok Bubon, West Aceh (Gazali, 2018) [14]. Erniati et al. (2022) [15] also reported that *Halimeda* sp. is spread over the west coast of Simeulue Island, Aceh, with a relative frequency of up to 9.22%. However, the application of green seaweed *Halimeda* sp., especially *H. tuna* species, is relatively low compared with that of brown and red seaweeds, especially the use of its bioactive compounds in the health sector. Furthermore, research on the antidiabetic activity of *H. tuna* has not been widely reported, especially the inhibition of α-amylase and α-glucosidase. Extraction using universal solvents and fractionation using the liquid–liquid partition method with several solvents with different levels of polarity can be carried out to show the antidiabetic potential of bioactive compounds from green seaweed *H. tuna*. In addition, the different types of seaweed and their growing location can affect the content and activity of bioactive compounds as antidiabetics. The purpose of this study was to investigate the antidiabetic activity of green seaweed *H. tuna* extracts from Lhok Bubon Coast, West Aceh District.

## 2. Results

### 2.1. Inhibitory Activity against α-Amylase

The inhibitory activity against α-amylase of the methanol extract of *H. tuna* can be seen in Figure 1. The percentage of inhibition of α-amylase by *H. tuna* extract and acarbose produced different results at each concentration. The higher the concentration used was, the higher the percentage of inhibitory activity against α-amylase obtained was. The highest inhibition percentage achieved with *H. tuna* methanol extract was at the concentration of 10 mg/mL (46.42 ± 1.00%), while the lowest inhibition was at the concentration of 0.625 mg/mL (17.10 ± 1.51%).

Figure 2 illustrates the inhibition of α-amylase by the ethyl acetate fraction of *H. tuna*, with the lowest concentration (0.16 mg/mL) having inhibitory activity of 24.88 ± 4.00% and the highest concentration (2.5 mg/mL) having inhibitory activity of 62.41 ± 1.92%. The inhibition of α-amylase by the water fraction of *H. tuna* can be seen in Figure 3; the lowest concentration (0.16 mg/mL) had inhibitory activity against α-amylase of 22.22 ± 0.48%, and the highest concentration (2.5 mg/mL) had inhibitory activity of 57.93 ± 1.55%. At concentrations of 0.63, 1.25, and 2.5 mg/mL, acarbose had higher inhibitory activity than the water fraction, but at concentrations of 0.16 and 0.31 mg/mL, the water fraction had higher inhibitory activity than acarbose.

### 2.2. Inhibitory Activity against α-Glucosidase

Figure 4 displays the inhibitory activity against α-glucosidase of *H. tuna* methanol extract. The results showed that methanol extract at a concentration of 10 mg/mL was able to inhibit an average of 97.75 ± 1.54% of enzyme activity, while acarbose at the same concentration inhibited 85.82 ± 3.80% of the activity of α-glucosidase. The ethyl acetate fraction of *H. tuna* extract at the concentration of 2.5 mg/mL had inhibitory activity of 98.06 ± 2.12%, while the lowest inhibition was at the concentration of 0.08 mg/mL (66.19 ± 3.33%), as shown in Figure 5. The inhibition of α-glucosidase by the water fraction of *H. tuna* is reported in Figure 6, showing that the highest concentration (2.5 mg/mL) had inhibitory activity of 61.10 ± 5.44%, while the lowest concentration (0.16 mg/mL) had inhibitory activity of 39.93 ± 2.57%.

Table 1 shows the IC_50_ inhibitory activity against α-amylase of the methanol extract, ethyl acetate fraction, and water fraction of *H. tuna,* and of acarbose. The ethyl acetate fraction (IC_50_ = 0.88 ± 0.20 mg/mL) showed inhibitory activity against α-amylase, which was not significantly different from that of acarbose (IC_50_ = 0.76 ± 0.04 mg/mL). However, the water fraction (IC_50_ = 1.50 ± 0.14 mg/mL) and the methanol extract (IC_50_ = 11.57 ± 0.37 mg/mL) had lower inhibitory activity than acarbose based on the IC_50_ values.

### 2.3. Phytochemical Test

The results of the phytochemical test in Table 2 show that the methanol extract of *H. tuna* contains bioactive compounds similar to those of the ethyl acetate fraction. However, the ethyl acetate fraction had a stronger yield tendency than the methanol extract in terms of color intensity and the resulting precipitate. The bioactive components of the methanol extract and ethyl acetate fraction of *H. tuna* include alkaloids, steroids, flavonoids, and phenol hydroquinone. Green seaweed *H. opuntia* obtained from the coast of West Aceh was extracted using three different solvents: ethanol, ethyl acetate, and n-hexane. The bioactive components found in ethanol solvent were alkaloids, flavonoids, phenols, tannins, and steroids; in ethyl acetate solvent, only phenol compounds were found, and in n-hexane solvent, there were none [16].

### 2.4. Identification of Active Compounds Using GC-MS

The inhibitors tested were the methanol extract and ethyl acetate fraction of *H. tuna.* The identification of *H. tuna* methanol extract compounds was intended as an initial screening to determine the compounds in the crude extract. Meanwhile, the identification of ethyl acetate fraction compounds was carried out to determine the active compounds playing a role in determining the high inhibitory activity of the ethyl acetate fraction of *H. tuna* against α-amylase and α-glucosidase. The compounds identified to have biological activity in the methanol extract of *H. tuna* are shown in Table 3. The results showed that *H. tuna* methanol extract compounds that have potential as antidiabetics were 1-Docosanol (5.03%), Neophytadiene (41.41%), Stigmasta-7,22-dien-3-ol, acetate,(3. beta.,5.alpha.,22E) (6.78%), Octadecanoic acid,2-oxo-, methyl ester (20.73%), and Phytol (42.43%). Table 4 presents the compounds with biological activity in the ethyl acetate fraction of *H. tuna*.

## 3. Discussion

The main principle in determining the type of solvent to be used in extraction is based on the solubility properties of the compounds to be extracted [32]. In this study, we used methanol as a solvent in the maceration process because methanol is a universal solvent capable of dissolving various compounds with different polarity levels. Extraction using methanol is able to extract the active components in the sample optimally so as to produce the highest antidiabetic activity [33].

For fractionation, in this study, we used the liquid–liquid partition method, referring to Basir et al.’s [34] method, with water and ethyl acetate (1:1) as solvents. The partition method for the isolation of secondary metabolites aims to classify compounds based on differences in their polarity levels. The choice of solvent used in this study was based on the nature of the secondary metabolites to be extracted. According to [35], the ethyl acetate compound is a semi-polar solvent that can dissolve semi-polar compounds on the cell wall. Therefore, the use of ethyl acetate solvent was expected to dissolve semi-polar active compounds in *H. tuna* extract, while water is a polar compound for dissolving compounds that are also polar. Semi-polar solvents are able to extract phenolic compounds, terpenoids, alkaloids, aglycones, and glycosides [35].

The inhibitory activity of *H. tuna* methanol extract was lower than that of acarbose when compared at each concentration. The results are in line with a study conducted by Chin et al. [36], who reported that *H. macroloba* seaweed had inhibitory activity against α-amylase at the highest concentration of 40 mg/mL. Pacheco et al. [37] also found that the acetone extract of *D. antarctica* seaweed at a concentration of 2 mg/mL was able to reduce α-amylase activity to 56.60 ± 2.00% and *Gelidium* sp. decreased the activity of α-amylase to 77.90 ± 2.10%. On the other hand, acarbose (concentration of 1000 µg/mL) decreased enzyme activity to 37.50 ± 0.40%. The percentages of inhibition by the ethyl acetate fraction at concentrations of 2.5 mg/mL and 1.25 mg/mL were lower than those of acarbose, while at lower concentrations, i.e., 0.63, 0.31, and 0.16 mg/mL, the ethyl acetate fraction had higher inhibitory activity than acarbose. The results are lower than the data reported in the research findings of Mohapatra et al. [38], who found that ethyl acetate extract of green seaweed *Ulva fasciata* had a percentage of inhibitory activity at a concentration of 100 µg/mL of 60.19 ± 2.24% and that the ethyl acetate fraction of *Gracilaria edulis* inhibited the activity of α-amylase at a concentration of 400 µg/mL by 64% [39]. Kumar et al. [40] reported that the aqueous extracts of seaweeds *U. lactuca, S. polycystum, G. edulis*, and *G. corticata* had an inhibitory effect on α-amylase. Green seaweed extracts of *Ulva lactuca* and *Ulva reticulata* had good inhibitory activity against α-amylase, 83.40 ± 2.50% and 89.10 ± 0.96%, respectively [41].

The results of this study agree with the research by Husni et al. [42], who reported that the extract of *S. hystrix* at a concentration of 10 mg/mL had inhibitory activity of 97.31 ± 1.46%, which is higher than the results of the study conducted by Chin et al. [32], where the aqueous extract of *H. macroloba* at a concentration of 40 mg/mL had a percentage of inhibition of the α-glucosidase enzyme of 80.94%.

Research conducted by Nguyen et al. [43] utilizing *Laurencia dendroidea* seaweed extracted using 80% methanol and fractionated with n-hexane, chloroform, ethyl acetate, and butanol showed that the ethyl acetate fraction had the strongest α-glucosidase inhibitory properties. This could have been due to the large content of polyphenols in the fraction. The mechanism of inhibition of α-glucosidase activity by polyphenolic compounds is thought to be the blocking of the active site of the diabetic enzyme by polyphenols, thereby changing the catalytic efficiency of the enzyme [44]. In addition, Husni et al. [42] reported that the ethyl acetate fraction of *S. hystrix* extract at a concentration of 100 μg/mL was able to inhibit the α-glucosidase enzyme by 84.47 ± 4.01%, while inhibition was 44.38 ± 7.50% at a concentration of 6.25 μg/mL. These results were higher than those of the study by Gunathilaka et al. [39], who reported that the inhibition of α-glucosidase by the ethyl acetate fraction of seaweed *Gracilaria edulis* varied from 6% (at the smallest concentration of 4.16 μg/mL) to 68% (at the concentration of 133.3 μg/mL).

The percentage of inhibition by the water fraction was lower than that of acarbose at concentrations from 2.5 mg/mL to 0.31 mg/mL, but at the concentration of 0.16 mg/mL, the percentages of inhibition by acarbose and water fraction were relatively similar. These results were higher than those of Sanger et al. [45], who reported that the inhibitory activity of methanol and hexane extracts as well as the chloroform and water fractions of *H. durvilae* at a concentration of 5 mg/mL were 18.71 ± 5.40%, 17.53 ± 3.55%, 33.9 ± 2.41%, and 44.56 ± 1.37%, respectively. The water fraction showed the highest percentage of inhibition at an IC_50_ of 4.34 ± 0.32 mg/mL, but lower than acarbose at the same concentration. In addition, the results obtained were also higher than those of the aqueous extract of *H. macroloba* (80.94%), showing a higher inhibitory activity against α-glucosidase than *T. conoides* (75.22%) at 40 mg/mL [36].

The inhibition activity of α-amylase by methanol extract was also higher than that by extracts of other seaweed species. Research by Payghami et al. [46], however, showed the same trend, wherein the IC_50_ value of inhibition by *Sargassum glaucescens* methanol extract was 8.90 ± 2.40 mg/mL, which was greater than that of inhibition by acarbose of 6.60 ± 2.10 mg/mL. Senthilkumar and Sudha [47] reported the IC_50_ values of inhibition of α-amylase by aqueous extracts of seaweed *U. lactuca, S. polycystum, G. edulis*, and *G. corticate* (67 µg/mL, 60 g/mL, 83 µg/mL, and 82 µg/mL). Mohapatra et al. [38] added that extraction using ethyl acetate of green seaweed *Ulva fasciata* had a strong inhibitory activity against α-amylase (IC_50_ = 69.12 g/mL) compared with the positive control, acarbose (IC_50_ = 49.34 µg/mL).

The active compounds from the methanol extract and the ethyl acetate fraction of *H. tuna* identified using the phytochemical tests are believed to play a role in providing antidiabetic activity. Surya et al. [48] reported that the phytochemical compounds identified were suspected to be bioactive compounds with the following characteristics: hypoglycemic effects, including alkaloids, flavonoids, steroids, glycosides, and protein tyrosine phosphatase-1B; inhibitory activity against advanced glycation end products; increased insulin secretion and sensitivity abilities; increased glucose absorption ability; and antioxidant ability [49]. Steroid compounds have strong antioxidant, hypoglycemic, and thyroid blocking properties [50]. Meanwhile, Lauro et al. [51] explained that the steroid derivative compound pregnenolone–dihydrotestosterone conjugate induces changes in glucose levels similar to glibenclamide. The antidiabetic activity of flavonoids acts on targets involved in type 2 diabetes mellitus, such as aldose reductase, α-glucosidase, and DPP-4 [52], and in insulin-dependent diabetes mellitus, the flavonoid compound quercetin has been reported to increase insulin release by increasing the regeneration of pancreatic islet cells [53]. Groups of phenols and flavonoids are those found in higher plants. The existing phenolic compounds with hydroxyl groups attached to aromatic rings are effective compounds with antioxidant and antibacterial activities, because these compounds reduce free radicals [54]. Hyperglycemic conditions can induce tissue damage and the production of free radicals, and the presence of phenolic compounds can limit oxidative stress due to the large production of free radicals. Therefore, antioxidant activity is closely related to reducing diabetes complications under hyperglycemic conditions. In addition, flavonoid compounds, among the phenolic compounds, have inhibitory activity against α-glucosidase [55].

The secondary metabolites of alkaloids, flavonoids, and terpenoids are naturally capable inhibitors of α-glucosidase. These secondary metabolites have been tested either in vitro or in vivo to reduce blood glucose levels [56]. Bioactive compounds that provide hypoglycemic effects include alkaloids, flavonoids, steroids, and glycosides [48]. Fatty acids are known to be able to inhibit the activity of carbohydrate-breaking enzymes, both α-amylase and α-glucosidase. Through the mechanism of competitive inhibition, fatty acid inhibitors are able to bind to the active site of the substrate, resulting in the active site of the enzyme being unable to react with the substrate [57]. Octadecanoic acid is suspected to be the cause of decreased blood glucose levels in STZ-induced diabetic rats [58].

Compound 1-Docosanol is one of the compounds identified and has antidiabetic activity. Jhong et al. [17] demonstrated the in vitro antidiabetic activity of docosanol and reported the ability of 1-Docosanol to inhibit digestive enzymes. In addition, the binding affinity of docosanol has been demonstrated in in silico studies. Janaki et al. [59] reported that the antidiabetic activity of *Fusinus nicobaricus* is indicated by the presence of one of the bioactive compounds, 1-Docosanol. Neophytadiene is a terpene group found in many plants, for example, in extracts of *Achlea ligustica*, which exhibits antiradical and antidiabetic potential with excellent yield [23]. A GC-MS analysis of red algae, *Centroceras clavulatum* (C. Agardh), revealed the presence of Neophytadiene and phytol, which function as analgesic, antidiabetic, antipyretic, and anti-inflammatory drugs [60]. Phytol has a potential role in the management of insulin resistance and metabolic disorders that accompany diabetes and obesity [25]. Phytol is an important diterpene with antimicrobial, antioxidant, and anticancer activities [61]. One of the compounds found in both the methanol extract and the ethyl acetate fraction of *H. tuna* that is thought to act as an antidiabetic is phytol. Elmazar et al. [25] reported that phytols have a potential role in the management of insulin resistance and metabolic disorders that accompany diabetes by activating retinoid X receptor (RXR) through its metabolites and modulating other factors associated with metabolic disorders. In addition, docking studies of phytanic acid molecules on two crystal structures of peroxisome proliferator-activated receptor nuclear receptors (PPARc)-binding protein and RXRa/PPARc heterodimers show that phytol acts by activating PPARs and heterodimerizing RXR with PPARc using phytanic acid. Stigmasta-7,22-dien-3-ol, acetate, (3.beta.,5.alpha.,22E) is a class of stigmasterol compounds. Poulose et al. [62] reported that stigmasterol compounds from seaweed *Gelidium spinosum* had better inhibitory activity than acarbose against α-amylase and α-glucosidase at the same concentration.

Table 4 shows that the Nonadecane compound has potential as an antidiabetic. Senarath et al. [31] reported that Nonadecane can be an antioxidant and antidiabetic compound due to the inhibition of the α-amylase enzyme in vitro. Eicosane and Nonadecane were reported to have antioxidant and antidiabetic potential [63]. The compound 14-.Beta.-H-pregna, a steroid group, is also estimated to have antidiabetic activity due to its hypolidemic activity [64]. Phytol compounds were also detected in the ethyl acetate fraction of *H. tuna* extract, and these compounds may have a role in the high antidiabetic activity of the ethyl acetate fraction. Phytol significantly suppresses the increase in postprandial blood glucose levels through the activation of AMPK, not AKT, in skeletal muscle and increases the abnormal pattern of insulin secretion in obese mice [65].

The octadecanoic acid compounds identified in this study may play a role in increasing the antidiabetic activity of the ethyl acetate fraction in inhibiting α-amylase and -glucosidase enzymes. Octadecanoic acid is suspected to be the cause of decreased blood glucose levels in STZ-induced diabetic rats [55]. These results correspond with the research of Osman and Hussein [66], who discovered that the compounds 9,12.5-octadecatrienoic acid methyl ester and 9,2-octadecadienoic acid methyl ester can protect the pancreas from changes in normality induced in diabetes. Wuttke et al. [21] also reported that hexadecanoic acid, octadecanoic acid, and eicosanoic acid have been shown to exhibit antidiabetic activity by influencing insulin secretion, insulin stimulation, and α-glucosidase inhibitors. Su et al. [57], in their research paper, explained that oleic acid and linoleic acid had the highest activity in inhibiting α-glucosidase, similar to acarbose in hydrolyzing starch, but had low activity in inhibiting α-amylase.

## 4. Materials and Methods

### 4.1. Sample Collection and Identification

Samples of green seaweed *H. tuna* were collected from the coast of Lhok Bubon, Samatiga Subdistrict, West Aceh District, Aceh Province. The samples were washed with fresh water to remove the adhering sand and dirt. The wet samples were then dried at room temperature. The wet and dry samples were sent to Universitas Gadjah Mada, Yogyakarta. Fresh seaweed samples were identified at Plant Systematics Laboratory, Faculty of Biology, Universitas Gadjah Mada, to determine the specific type. Dry samples were cut into 1 cm pieces using scissors. Seaweed was weighed and stored at −20 °C.

### 4.2. Extraction of Seaweed

The extraction of green seaweed *H. tuna* was performed following Azizi et al.’s [67] method with modifications. The dried *H. tuna* samples were dissolved in methanol (1:8). The extraction process was carried out for 24 h at room temperature. Next, the samples were filtered using Whatman filter paper No. 42. The sample filtrate was evaporated using a rotary evaporator (40 °C, 60 rpm), followed by nitrogen evaporation. The extract was dried using a freeze-dyer for 24 h and stored at −20 °C.

### 4.3. Liquid–Liquid Partition

The liquid–liquid partition of green seaweed *H. tuna* extract was carried out based on the method of Basir et al. [34] with slight modifications. In the liquid–liquid partition method, ethyl acetate and water are used in a ratio of 1:1 (*v*/*v*). The dry extract was dissolved in ethyl acetate and water (1:1, *v*/*v*) into a separating funnel. The mixture was shaken slowly for 10 min and allowed to stand until the water and ethyl acetate fractions separated. The ethyl acetate fraction (top) was concentrated in a water bath at 50 °C, and the water fraction (bottom) was concentrated in a water bath at 60–80 °C. The resulting dry yield was then stored at −20 °C.

### 4.4. Inhibitory Activity against α-Amylase

The inhibitory activity against α-amylase was assessed by referring to the method of Husni et al. [42] with minor modifications. The samples used were methanol extract, ethyl acetate fraction, water fraction, and acarbose as a control. The *S1* solution was prepared by inserting 25 µL of the sample dissolved in dimethyl sulfoxide (DMSO) into the microplate, followed by adding α-amylase at 13 U/mL (25 µL). The mixture was incubated for 10 min at 37 °C. Next, the mixture was combined with 1% starch as the substrate (25 µL) and re-incubated at 37 °C for 10 min. After the incubation process, the mixture was supplemented with 50 µL of 3,5-dinitrosalicylic acid (DNS) color reagent and heated in boiling water for 5–10 min to stop the reaction. The mixture was cooled at room temperature and transferred into a cuvette filled with 500 mL of distilled water. The *S0* solution was prepared similarly, but the addition of α-amylase was replaced with 25 µL of phosphate buffer at pH 7. While the preparation of solutions *K* and *B* was also performed in the same way as *S1* and *S0*, the sample was replaced with phosphate buffer. Each test system was repeated three times. Absorbance was measured with a spectrophotometer at a wavelength of 540 nm. Inhibition was calculated based on the following formula:$$\mathrm{Inhibition}\;(\%)=\frac{\left(K-B\right)-\left(S1-S0\right)}{\left(K-B\right)}\;\times \;100$$
where *K* = control with enzyme addition; *B* = control without enzyme addition; *S1* = sample with addition of enzyme; and *S0* = sample without addition of enzyme.

### 4.5. Inhibitory Activity against α-Glucosidase

The inhibitory activity against α-glucosidase was assessed by referring to the method of Azizi et al. [68] with slight modifications. The samples used were methanol extract, ethyl acetate fraction, water fraction, and acarbose as a control. The preparation of the *S1* solution was carried out by inserting 50 µL of phosphate buffer into the microplate before adding 25 µL of 0.5 mM p-Nitrophenyl-α-D-glucopyranoside (p-NPG) substrate. The mixture was mixed with 25 µL of the sample and dissolved in dimethyl sulfoxide (DMSO) and 25 µL of 0.2 U/mL α-glucosidase. The mixture was incubated for 30 min at 37 °C. The reaction was stopped by adding 100 µL of Na_2_CO_3_ at 0.2 M. The *S0* solution was prepared in the same way, but α-glucosidase was replaced with 75 µL of phosphate buffer at pH 7. Meanwhile, the *K* and *B* solutions were prepared similarly to *S1* and *S0*, where the sample was replaced with phosphate buffer. Each test system was repeated three times. The inhibitory activity was measured with the amount of p-nitrophenol produced by measuring its absorbance using an ELISA microplate reader at 405 nm. Inhibition was calculated based on the following formula:$$\mathrm{Inhibition}\;(\%)=\frac{\left(K-B\right)-\left(S1-S0\right)}{\left(K-B\right)}\;\times \;100$$
where *K* = control with enzyme addition; *B* = control without enzyme addition; *S1* = sample with addition of enzyme; and *S0* = sample without addition of enzyme.

### 4.6. Phytochemical Analysis

Phytochemical analysis is a qualitative analysis to determine the content of bioactive components in the extract. The presence of flavonoids, steroids, triterpenoids, tannins, alkaloids, phenol hydroquinone, and saponins in the methanol extract and the ethyl acetate fraction of *H. tuna* were investigated. The color intensity or precipitate formation was used as an analytical response for this analysis.

#### 4.6.1. Flavonoid Test

The flavonoid test was performed according to the method of Devi [68] with some modifications. The sample (2 drops) was put into a test tube; then, 2–4 drops of 10% NaOH solution were added. A color change from yellow to brownish yellow indicated that the sample contained flavonoids.

#### 4.6.2. Steroid and Triterpenoid Test

Steroids and triterpenoids were tested based on the method of Widiowati et al. [69] with modifications. The extract (25 mg) was added to chloroform (1:1) on a drip plate and allowed to dry; then, 5 drops of anhydrous acetic acid CH_3_COOH were added and stirred to obtain a homogeneous mixture. Then, concentrated acetic acid (H_2_SO_4_) was added to the mixture. The results showed the content of triterpenoids, indicated by the formation of red or purple color, and the content of steroids, indicated by the formation of green or blue color.

#### 4.6.3. Tannin Test

The modified method of Widiowati et al. [69] was used for the tannin test on H. tuna extract. The sample (25 mg) was dissolved with 10 mL of hot aquabides, and the solution was then filtered. The filtrate (5 mL) was put into a test tube, and 2 drops of 1% FeCl_3_ were added. The presence of tannins was indicated by blackish color.

#### 4.6.4. Alkaloid Test

The modified method of Widiowati et al. [69] was applied for alkaloid testing. The extract (25 mg) was put into a vial, supplemented with 5 drops of NH_3_ and 5 mL of CHCl_3_, and shaken. Later, 5 drops of 2 M H_2_SO_4_ were added and stirred. The solution was then allowed to stand to form a layer. The first layer formed was taken and divided into three test tubes. Furthermore, in each test tube, reagents were added successively, namely, Dragendorff’s, Mayer’s, and Wagner’s reagents. The test results with Dragendorff’s reagent are positive for alkaloids if a red or orange precipitate forms, while Mayer’s reagent indicates that the sample is positive for alkaloids if a yellowish white precipitate forms, and if using Wagner’s reagent, the presence of alkaloids can be seen when a brown precipitate forms.

#### 4.6.5. Saponin Test

Saponin testing was performed using the modified method of Harborne [35]. The sample (25 mg) was dissolved with hot water (1:1) into a vial. If stable foam formed after the vial was shaken and allowed to stand for 30 min and did not disappear after 2 drops of 2 N HCl solution was added, this indicated that the sample contained saponins.

#### 4.6.6. Phenol Hydroquinone Test

Phenol hydroquinone was tested according to the modified method of Harborne [35]. The sample (25 mg) was put into a test tube. The sample was then mixed with 2 mL of 70% ethanol and supplemented with 2 drops of 5% FeCl_3_. Sample content of phenol hydroquinone was indicated by the change in color to green/blue to red.

### 4.7. Identification of Active Compounds Using GC-MS

Compounds were identified using the gas chromatography–mass spectrometry (GC-MS) method. In GC-MS, the analyte is separated using gas chromatography, and its identity is confirmed using mass spectrophotometry techniques [70]. The sample was first dissolved in 50 µL of organic solvent. A sample of 10 µL was injected in the injection port at 290 °C. The volatilized sample was carried by helium at a flow rate of 1 mL/min through a gas chromatography column. The temperature at the time of injection was 80 °C and increased by 10 °C per minute, with a final temperature of 300 °C (43 min) [70]. The detection of compounds took place in a mass spectrometry system with the mechanism of crashing or bombarding compounds with electrons to form ionized molecules and record fragmentation patterns. The fragmented mass components were compared with WILEY and NIST standard reference data, as indicated by the percentage similarity index (SI) [38].

### 4.8. Data Analysis

The data obtained in this study were in the form of extract concentration versus percent inhibition of α-amylase and α-glucosidase. The data were then plotted to generate a regression equation. The IC_50_ activity values of *H. tuna* extract and its fractions against α-amylase and α-glucosidase were obtained using the regression equation. The percent inhibition value of each sample and the IC_50_ value were tested statistically with one-way ANOVA. If the treatment was significant, Tukey’s HSD test was employed. Meanwhile, the percent yield produced was calculated with a T-test for unpaired data using (SPSS) with a 95% confidence interval.

## 5. Conclusions

Extracts and fractions of *H. tuna* showed inhibitory activity against α-amylase and α-glucosidase. The ethyl acetate fraction (IC_50_ = 0.87 ± 0.20 mg/mL) showed inhibitory activity against α-amylase that was close to that of acarbose (IC_50_ = 0.76 ± 0.04 mg/mL) but higher than that of the water fraction (IC_50_ = 1.50 ± 0.13 mg/mL) and methanol extract (IC_50_ = 11.57 ± 0.37 mg/mL). Methanol extract (IC_50_ = 0.05 ± 0.01 mg/mL) and ethyl acetate fraction (IC_50_ = 0.01 ± 0.00 mg/mL) showed higher inhibitory activity against α-glucosidase than acarbose (IC_50_ = 0.27 ± 0.13 mg/mL), but the water fraction (IC_50_ = 0.55 ± 0.12 mg/mL) showed lower activity. The methanol extract and ethyl acetate fraction of *H. tuna* contained secondary metabolite components: alkaloids, steroids, flavonoids, and phenol hydroquinone. Compounds believed to play a role in the inhibition of α-amylase and α-glucosidase were found in methanol extract, namely, 1-Docosanol, Neophytadiene, Stigmasta-7,22-dien-3-ol,acetate,(3.beta.,5. alpha.,22E), Octadecanoic acid,2-oxo-,methyl ester, and Phytol. Meanwhile, the ethyl acetate fraction contained n-Nonadecane, Phytol, Butyl Ester, 14-Beta.-H-Pregna, Octadecenoic acid, and Oleic acid. However, until now, it is not known exactly which compound acts as an antidiabetic in H. tuna. Therefore, further isolation and purification are necessary to obtain antidiabetic compounds from seaweed.

## Figures and Tables

**Figure 1 plants-12-00393-f001:**
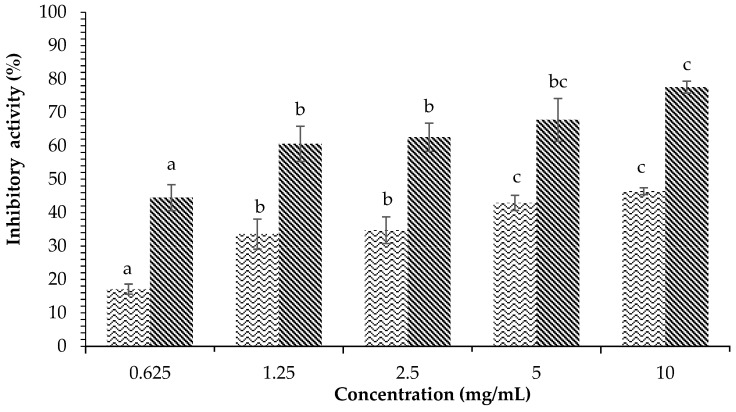
Effects of concentration of *H. tuna* methanol extract (
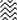
) and acarbose (
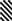
) on α-amylase inhibition. Each value is expressed as mean ± SD in the triplicate experiment. Values (a–c) with different alphabet letters indicate significant differences among treatments at *p* < 0.05, which was analyzed using Tukey’s HSD.

**Figure 2 plants-12-00393-f002:**
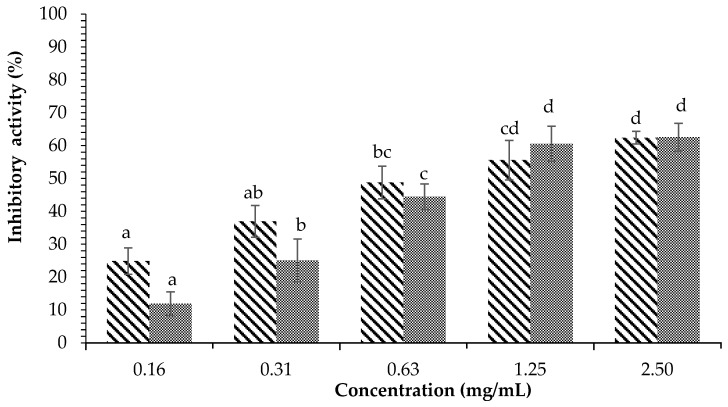
Effects of concentration of ethyl acetate fraction of *H. tuna* (
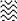
) and acarbose (
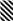
) on α-amylase inhibition. Each value is expressed as mean ± SD in the triplicate experiment. Values (a–d) with different alphabet letters indicate significant differences among treatments at *p* < 0.05, which was analyzed using Tukey’s HSD.

**Figure 3 plants-12-00393-f003:**
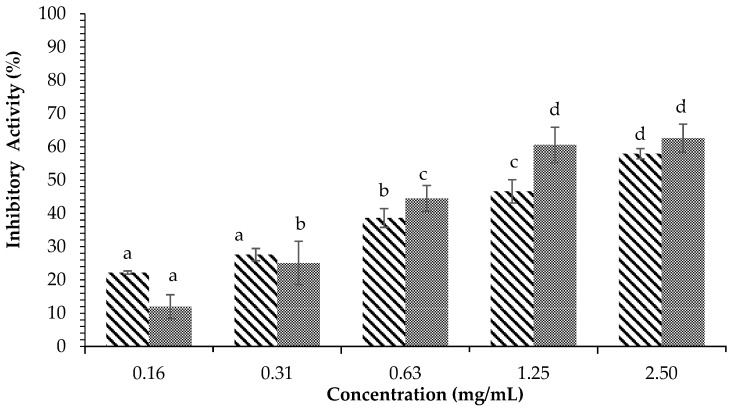
Effects of concentration of water fraction of *H. tuna* (
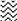
) and Acarbose (
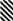
) on α-amylase inhibition. Each value is expressed as mean ± SD in the triplicate experiment. Values (a–d) with different alphabet letters indicate significant differences among treatments at *p* < 0.05, which was analyzed using Tukey’s HSD.

**Figure 4 plants-12-00393-f004:**
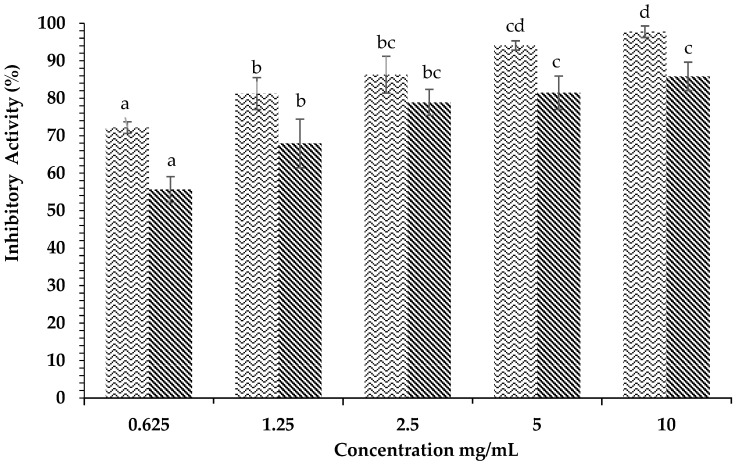
Effects of concentration of methanol extract of *H. tuna* (
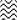
) and Acarbose (
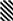
) on α-glucosidase inhibition. Each value is expressed as mean ± SD in the triplicate experiment. Values (a–d) with different alphabet letters indicate significant differences among treatments at *p* < 0.05, which was analyzed using Tukey’s HSD.

**Figure 5 plants-12-00393-f005:**
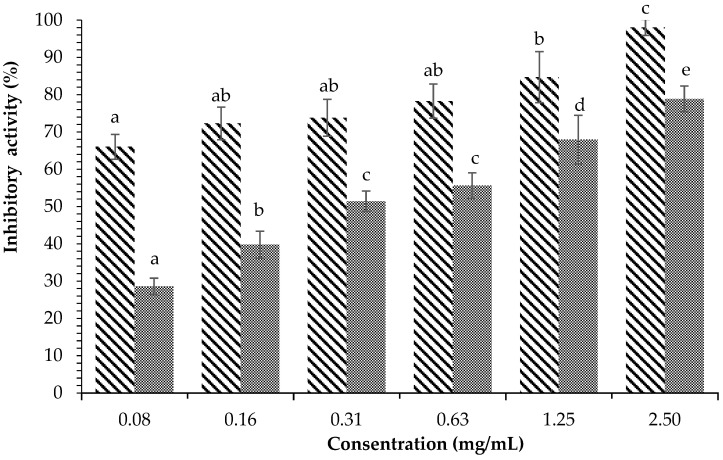
Effects of concentration of ethyl acetate fraction of *H. tuna* (
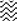
) and Acarbose (
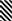
) on α-glucosidase inhibition. Each value is expressed as mean ± SD in the triplicate experiment. Values (a–e) with different alphabet letters indicate significant differences among treatments at *p* < 0.05, which was analyzed using Tukey’s HSD.

**Figure 6 plants-12-00393-f006:**
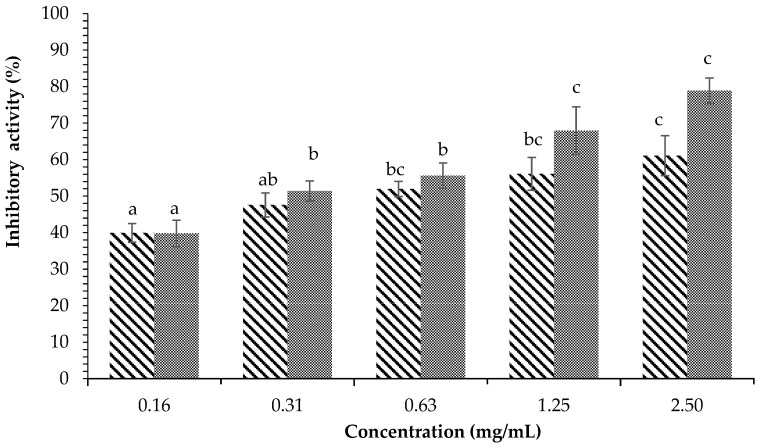
Effects of concentration of water fraction of *H. tuna* (
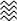
) and Acarbose (
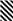
) on α-glucosidase inhibition. Each value is expressed as mean ± SD in the triplicate experiment. Values (a–c) with different alphabet letters indicate significant differences among treatments at *p* < 0.05, which was analyzed using Tukey’s HSD.

**Table 1 plants-12-00393-t001:** Inhibitory activity (IC_50_) of *H. tuna* methanol extract, ethyl acetate fraction, water fraction, and acarbose against α-amylase and α-glucosidase.

Inhibitor	α-Amylase (mg/mL)	α-Glucosidase (mg/mL)
Methanol extract	11.58 ± 0.38 ^a^	0.05 ± 0.01 ^ab^
Water fraction	1.50 ± 0.13 ^b^	0.55 ± 0.12 ^c^
Ethyl acetate fraction	0.87 ± 0.20 ^c^	0.01 ± 0.00 ^a^
Acarbose	0.76 ± 0.04 ^c^	0.27 ± 0.13 ^b^

Each value is expressed as mean ± SD in the triplicate experiment. Values (a–c) with different alphabet letters indicate significant differences among treatments at *p* < 0.05, which was analyzed using Tukey’s HSD.

**Table 2 plants-12-00393-t002:** Phytochemical test results of methanol extract and ethyl acetate fraction of *H. tuna*.

Test	Methanol Extract	Ethyl Acetate Fraction	Standard (Color)
Alkaloids			
Dragendorff	+	++	Red or orange precipitate
Mayer	-	-	Yellowish precipitate
Wagner	-	-	Brown precipitate
Steroids	+	++	Green/blue
Triterpenoids	-	-	Red/purple
Saponins	-	-	Stable foam forms
Flavonoids	++	+	Yellow-orange
Phenol hydroquinone	+	+	Green/blue to red
Tannins	-	-	Dark blue color

Description: - = Not detected; + = Weak; ++ = Strong.

**Table 3 plants-12-00393-t003:** Biological activity of compounds in methanol extract of *H. tuna*.

No.	RT	% Area	Compound	Activity
1	18.010	5.03	1-Docosanol	Antidiabetic activity [17]
2	20.441	27.41	2-Nonanol, 5-ethyl-	Anticancer activity [18]
3	20.833	6.78	Stigmasta-7,22-dien-3-ol, acetate, (3.beta., 5.alpha.,22E)-	Antiulcerogenic and antithrombotic activities [19]
4	21.065	20.73	Octadecanoic acid, 2-oxo-, methyl ester	Antibacterial [20] and antidiabetic activities [21]
6	13.516	41.41	Neophytadiene	Antipyretic, analgesic and anti-inflammatory, antimicrobial, antioxidant [22], and antidiabetic activities [23]
7	13.520	42.43	Phytol	Anxiolytic activity, metabolic modulation, cytotoxic activity, antioxidant activity, induces apoptosis, antinociceptive activity, anti-inflammatory activity, immune modulation, antimicrobial effect [24], and antidiabetic activity [22,25]

**Table 4 plants-12-00393-t004:** Biological activity of compounds in ethyl acetate fraction of *H. tuna* extract.

No.	RT	% Area	Compound	Activity
1	19.057	3.06	Octadecyl vinyl ether	Antisepsis activity [26]
2	19.380	7.78	n-Tetratetracontane	Antioxidant, anti-inflammatory, antibacterial, and antiulcerogenic activities [27]
3	19.508	8.32	3-Ethyl-5-(2′-ethylbutyl) octadecane	Antioxidant effect and anti-inflammatory activity [28]
4	19.555	11.30	n-Dotriacontane	Anticonvulsant activity [29], antioxidant activity, and stomach cramp reliever [30]
5	23.220	69.53	n-Nonadecane	Antidiabetic activity [31]

## Data Availability

The data used to support the findings of this study have been deposited in the Universitas Gadjah Mada repository (http://etd.repository.ugm.ac.id/penelitian/detail/212381).

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
