# Peer review of "In Vitro α-Amylase and α-Glucosidase Inhibitory Activity of Green Seaweed Halimeda tuna Extract from the Coast of Lhok Bubon, Aceh"

_plants, 2023, doi:10.3390/plants12020393_

Round 1
Reviewer 1 Report
The authors' exploratory study is very interesting. The following are some of my concerns.
(1) The authors should examine the Inhibitory activity against α-glucosidase using mice.
(2) What is the evidence for the concentration settings used by the authors in their comparison of acarbose and extracts? Are the concentrations of extract and acarbose the same?
(3) The graph figures are poorly visible. The numbers on the graph should be removed.
Author Response
Point 1: The authors should examine the Inhibitory activity against α-glucosidase using mice.
Response 1: Experiments on anti-diabetic activity can be carried out in vitro and/or in vivo (using mice). We have tested the anti-diabetic activity of Na-alginate from Turbinaria ornata seaweed using mice and it has been published in the Journal of Technology (Sciences & Engineering) 78:4–2 (2016) 7–14. In the manuscript that we are currently writing, we report the results of in vitro anti-diabetic studies, while we will carry out in vivo tests for the next research.
Point 2: What is the evidence for the concentration settings used by the authors in their comparison of acarbose and extracts? Are the concentrations of extract and acarbose the same?
Response 2: In setting concentrations, we use published journal (Azizi et al., 2019. Inhibitory activity of Sargassum hystrix extract and its methanolic fractions on inhibiting α-glucosidase activity. Indonesian Journal of Pharmacy 30(1): 35-42.) as a reference. Yes, the extract and acarbose was used in same concentration.
Point 3: The graph figures are poorly visible. The numbers on the graph should be removed.
Response 3: Thank you. The graph figures already revised.

Reviewer 2 Report
1. Review mdpi paper
-Proof reading and language improvement of manuscript essential; errenous sentences for eg:
I. acetate fraction were was n-Nonadecane, Phytol, Butyl ester, 14-.Beta.-H-pregna, Octadecenoic acid, line 32
II. have been widely used to treat type 2 diabetes patients, such as vomiting, hepatic impairment, and dizziness. Therefore, there is a need.., line 52 & 53
III. Avoid sentences such as ‘The results obtained are the same as the research conducted by Husni et al.’ As the results are not same as the study was conducted on different seaweed.
2. Why were the extract bio-actives not quantified and do the authors think using pure standards of the reported antidiabetic compounds found in the extracts studied would help in establishing the bio-activity of these compounds in a dose dependent response curve.
3. Discuss possible role of peptides from seaweed in inhibition of the implicated enzymes?
4. Comment on the toxicity of compounds such as nonadecane or 1-docosanol.
Author Response
Point 1: Review mdpi paper. -Proof reading and language improvement of manuscript essential; errenous sentences for eg: I. acetate fraction were was n-Nonadecane, Phytol, Butyl ester, 14-.Beta.-H-pregna, Octadecenoic acid, line 32. II. have been widely used to treat type 2 diabetes patients, such as vomiting, hepatic impairment, and dizziness. Therefore, there is a need.., line 52 & 53. III. Avoid sentences such as ‘The results obtained are the same as the research conducted by Husni et al.’ As the results are not same as the study was conducted on different seaweed.
Response 1: The manuscript has been proofread by the Translating, Proofreading, and Data Analysis "Read-a-ble" Institute.
Point 2: Why were the extract bio-actives not quantified and do the authors think using pure standards of the reported antidiabetic compounds found in the extracts studied would help in establishing the bio-activity of these compounds in a dose dependent response curve.
Response 2: The yields for methanol extract, ethyl acetate fraction and water fraction were 0.13±0.03%, 0.03±0.00%, and 0.01±0.00%, respectively. In this study we used Acarbose as a comparison. Acarbose is a drug commonly used to lower blood sugar levels in people with type 2 diabetes.
Point 3: Discuss possible role of peptides from seaweed in inhibition of the implicated enzymes?
Response 3: Research on α-amylase inhibitory activity by peptides has been reported by Admassu et al. (J. Agric. Food Chem. 2018, 66, 4872−4882). In that study, Red Seaweed (Porphyra spp) was hydrolyzed enzymatically and two peptides (Gly-Gly-Ser-Lys and Glu-Leu-Ser) were obtained which had α-amylase inhibitory activity.
Point 4: Comment on the toxicity of compounds such as nonadecane or 1-docosanol.
Response 4: Nanodecane: Not a hazardous substance or mixture according to Regulation (EC) No. 1272/2008. Docosanol: This product is for research use - Not for human or veterinary diagnostic or therapeutic use.

Reviewer 3 Report
The manuscript presents determinations regarding the inhibitory activity of fitochemical compounds from aqueous, methanolic and ethylacetate extracts from green seaweed Halimeda tuna on α-amilase and α-glucosidase.
Please take the following remarks into consideration:
I sugest explaining the details about the choice of the three solvents (water, methanol, ethyl acetate) in the Discussions section
I recomand improving the conclusions regarding the activity of the fitochemical compounds identified by GC-MS and you could specify future reasearch directions in this section as well.
Author Response
Point 1: I sugest explaining the details about the choice of the three solvents (water, methanol, ethyl acetate) in the Discussions section
Response 1:
The main principle in determining the type of solvent to be used in extraction is based on the solubility properties of the compounds to be extracted (Kemit et al., 2016). In this study using methanol as a solvent in the maceration process because methanol is a universal solvent capable of dissolving various compounds with different polarity levels. Extraction using methanol is able to extract the active components in the sample optimally so as to produce the highest antidiabetic activity (Iwai, 2008).
The fractionation in this study used the liquid-liquid partition method referring to the Basir et al. (2017) method with water and ethyl acetate (1:1) as solvents. The partition method in the isolation of secondary metabolites aims to classify compounds based on differences in their polarity levels. The choice of solvent used in this study was based on the nature of the secondary metabolites to be extracted. According to (Harborne, 1987) ethyl acetate compound is a semi-polar solvent that can dissolve semi-polar compounds on the cell wall. Therefore, the use of ethyl acetate solvent is expected to dissolve the semi-polar active compound in H. tuna extract. While water is a polar compound in dissolving compounds that are also polar. Semi-polar solvents are able to extract phenolic compounds, terpenoids, alkaloids, aglycones and glycosides (Harborne, 1987).
Point 2: I recomand improving the conclusions regarding the activity of the fitochemical compounds identified by GC-MS and you could specify future reasearch directions in this section as well.
Response 2: Conclusion: Extracts and fractions of H. tuna showed inhibitory activities against α-amylase and α-glucosidase. The ethyl acetate fraction (IC50 = 0.87 ± 0.20 mg/mL) showed an inhibitory activity against the α-amylase which was close to acarbose (IC50 = 0.76 ± 0.04 mg/mL) but higher than the water fraction (IC50 = 1.50 ± 0.13 mg/mL) and methanol extract (IC50 = 11.57 ± 0.37 mg/mL). Methanol extract (IC50 = 0.05 ± 0.01 mg/mL) and ethyl acetate fraction (IC50 = 0.01 ± 0.00 mg/mL) showed higher inhibitory activity against α-glucosidase than acarbose (IC50 = 0.27 ± 0.13 mg/mL) but water fraction (IC50 = 0.55 ± 0.12 mg/mL) showed lower activity. The methanol extract and the ethyl acetate fraction of H. tuna contain secondary metabolite components: alkaloids, steroids, flavonoids and phenol hydroquinone. The compounds believed to play a role in inhibition of α-amylase and α-glucosidase were found in compounds in methanol extract, namely 1-Docosanol, Neophytadiene, Stigmasta-7,22-dien-3-ol,acetate,(3.beta.,5. alpha.,22E), Octadecanoic acid,2-oxo-,methyl ester and Phytol. Meanwhile, the ethyl acetate fraction was n-Nonadecane, Phytol, Butyl Ester, 14-Beta.-H-Pregna, Octadecenoic acid, and Oleic acid. However, until now it is not known exactly which compound acts as an antidiabetic from H. tuna. Therefore, further isolation and purification is necessary to obtain the antidiabetic compound from the seaweed.

Round 2
Reviewer 1 Report
To accept the authors' manuscripts, the authors need to identify the chemical compounds in the extract and clarify their mechanism of inhibitory activity, or demonstrate the biological activity of the extract.
Author Response
One of the compounds found in both the methanol extract and the ethyl acetate fraction of H. tuna which is thought to act as an antidiabetic is phytol. Elmazar et al. (2013) reported that phytols have a potential role in the management of insulin resistance and metabolic disorders that accompany diabetes, by activating the retinoid X receptor (RXR) through its metabolites, and modulating other factors associated with metabolic disorders. In addition, docking studies of phytanic acid molecules on two crystal structures of peroxisome proliferator-activated receptor nuclear receptors (PPARc)-binding protein, and RXRa/PPARc heterodimers show that phytol acts by activating PPARs and heterodimerizing RXR with PPARc by phytanic acid.

Reviewer 3 Report
The marine biota contain nutrients and secondary metabolites with health benefits.. It is important to continue research on them.